# Fabric-Reinforced Cementitious Matrix (FRCM) Carbon Yarns with Different Surface Treatments Embedded in a Cementitious Mortar: Mechanical and Durability Studies

**DOI:** 10.3390/ma15113927

**Published:** 2022-05-31

**Authors:** Francesca Bompadre, Jacopo Donnini

**Affiliations:** Department of Materials, Environmental Sciences and Urban Planning (SIMAU), Marche Polytechnic University, 60131 Ancona, Italy; f.bompadre@univpm.it

**Keywords:** Fabric-Reinforced Cementitious Matrix (FRCM), carbon yarn, carbon fibers, surface treatments, coating, interface, bond, environmental exposure, durability

## Abstract

Nowadays, FRCM systems are increasingly used for the strengthening and retrofitting of existing masonry and reinforced concrete structures. Their effectiveness strongly depends on the bond that develops at the interface between multifilament yarns, which constitute the reinforcing fabric, and the inorganic matrix. It is well known that fabric yarns, especially when constituted by dry carbon fibers, have poor chemical-physical compatibility with inorganic matrices. For this reason, many efforts are being concentrated on trying to improve the interface compatibility by using different surface treatments on multifilament yarns. In this paper, three different surface treatments have been considered. The first two involve yarn pre-impregnation with flexible epoxy resin or nano-silica coating, while the third one involves a fiber oxidation process. Uniaxial tensile tests were carried out on single carbon yarns to evaluate tensile strength, elastic modulus and ultimate strain before and after surface treatments, and also after yarn exposure to accelerated artificial aging conditions (1000 h in saline or alkaline solutions at 40 °C), to evaluate their long-term behavior in aggressive environments. Pull-out tests on single carbon yarns embedded in a cementitious mortar were also carried out, under normal environmental conditions and after artificial exposure. Epoxy proved to be the most effective treatment, by increasing the yarn tensile strength of 34% and the pull-out load of 138%, followed by nano-silica (+9%; +40%). All surface treatments were shown to remain effective even after artificial environmental exposures, with a maximum reduction of yarn tensile strength of about 13%.

## 1. Introduction

Fabric-Reinforced Cementitious Matrix (FRCM), also known in the international literature as Textile-Reinforced Mortar (TRM), is a new class of composite material that has generated considerable interest as a strengthening technique for concrete and masonry structures. FRCMs are constituted by open grids of perpendicularly connected multifilament yarns (made of carbon, glass, aramid, basalt or PBO fibers), which are applied on concrete or masonry structural elements through lime or cement-based matrices [1,2,3,4]. Although the use of FRCM systems as externally bonded reinforcement is nowadays common practice in civil engineering, there are still some issues that need to be addressed, such as the modest adhesion at the interface between the fabric reinforcement, made of multifilament yarns, and the inorganic matrix.

Experimental studies on FRCM systems showed that the main failure mode is due to debonding and slippage of the fabric yarns within the inorganic matrix [5,6]. Slippage of multifilament yarns within the matrix is emphasized in the case of yarns made up of carbon fibers [7]. The absence of hydrophilic groups on the carbon chain indeed results in a relatively poor chemical compatibility between the surface of carbon fibers and inorganic matrices [8]. Moreover, the inorganic matrix (usually cement-based) is not able to fully penetrate between the filaments of the yarn due to its high viscosity, thus allowing the innermost filaments to slip over each other, showing the so-called telescopic behavior [9].

Several solutions have been proposed to improve the bond at the fiber-to-matrix interface; some of them directly modify the fiber surface by means of chemical-physical reactions (e.g., fiber oxidation [8,10]), while others provide for partial or complete pre-impregnation of the fibers (usually with organic polymers). Pre-impregnation of the fibers with organic coatings has been reported as a promising technique to improve the mechanical performance of FRCM systems by many studies [11,12,13,14,15,16]. Donnini et al. found that the use of epoxy coatings on carbon fabrics [17] or yarns [7] is very promising in increasing the bond at the interface with the inorganic matrix, depending on the level of pre-impregnation. Signorini et al. investigated the effect of epoxy resin viscosity on the mechanical properties of pre-impregnated FRCM systems, showing that epoxy can penetrate inside the yarn and prevent telescopic failure [18].

In general, the main consequence of pre-impregnation with organic coatings is that the inorganic matrix is prevented from penetrating within the filaments of the yarn, and the interface bond is no longer dependent on the matrix’s ability to wet the single filaments, but rather on the interaction between the coating and the matrix. Moreover, when the single filaments are embedded in an organic coating, a higher number of filaments are engaged in the stress-transfer mechanisms, thus improving the yarn tensile strength and also leading to a change in the failure mode (usually from fabric slippage to fabric breakage) [19]. However, the main disadvantage of using organic coatings is the reduced ability of the composite system to withstand high temperatures [20,21,22].

Alternatively, the use of inorganic coatings has also been investigated in order to overcome the issue of polymeric matrices subject to high temperatures [23,24,25]. The use of inorganic coatings, usually based on silica, cement or other nano-particles, allows for a stronger fiber–matrix interaction, due to the reaction and chemical bonds that can develop between the coating and the inorganic matrix, thus providing better adhesion at the fiber-to-matrix interface [25,26,27]. The interfacial behavior between nano-silica modified carbon fibers and cementitious matrices has been recently studied by Li et al. [28], showing that the interfacial adhesion at the fiber–matrix interface is improved, compared to untreated fibers, due to the formation of C-S-H gel in the vicinity of the fiber surface. The effects of silica nano-coatings on carbon fibers have also been investigated by Signorini et al. [29], showing a good improvement in the performance of the composite, even if lower than those obtained with polymeric coatings.

The durability of FRCM systems is also a very topical issue [13,30,31,32,33]. Carbon fibers have high resistance to chemical attacks and their properties remain almost unchanged when subjected to artificial aging conditions [34,35]. However, when the fibers are coated or modified with other surface treatments, the mechanical properties and durability of the yarns could be compromised.

The objective of this study is to evaluate the effectiveness of three different surface treatments applied to multifilament carbon yarns, to improve their mechanical performance and chemical-physical compatibility with cement-based matrices. The treatments investigated consist of epoxy pre-impregnation, nano-silica coating and sonication through an oxidative solution. The effectiveness of the treatments was evaluated both in terms of the tensile properties of the yarns (uniaxial tensile tests on single yarns) and the bond with a cement-based mortar (pull-out tests). Moreover, mechanical properties have also been evaluated after exposing the specimens under different artificial aging conditions, to verify their effectiveness even in aggressive environments (Figure 1).

## 2. Materials and Methods

### 2.1. Materials

Two different types of specimens were manufactured to evaluate the effectiveness of different surface treatments on the tensile properties of single carbon yarns and on the adhesion with an inorganic cement-based mortar (as schematically reported in Figure 1). The carbon yarns used in the experiments have been taken from a commercially available bidirectional dry carbon fabric. The 24 K carbon yarn has a cross-sectional area equal to 1.04 mm^2^, as reported by the manufacturer, according to ISO 527-4,5: 2021 [36]. The preparation of carbon yarns for tensile tests, test procedure and mechanical properties are reported in the experimental results section.

Pull-out specimens were manufactured by embedding single carbon yarns in a cube of cement-based mortar (side of 40 mm), the mix proportions and mechanical properties of which have been reported in Table 1. Compressive and flexural strength of the mortar were evaluated on prismatic specimens (40 × 40 × 160 mm^3^) after 28 days of curing in laboratory conditions (20 ± 2 °C, RH = 70%), according to UNI EN 1015-11 [37].

### 2.2. Surface Treatments

Three different surface treatments were employed. The first one consists of the application of a highly flexible two-component epoxy resin (C-E), the mechanical properties of which are reported in Table 2. The yarns were fully impregnated with the epoxy by means of a plastic spatula, then cured at 60 °C for 24 h.

The second treatment involved the application of a nano-silica coating (C-NS). Carbon yarns were immersed in a nano-silica dispersion under stirring for 15 min and then dried at room temperature. The nano-silica dispersion was obtained using the sol-gel method by adding an acidic solution (distilled water: 65% nitric acid in the molar ratio 1:0.032) to a 98% tetraethyl orthosilicate (TEOS) by Evonik, ethanol (analytical grade) solution as described in [38].

Finally, the third treatment (C-Ox) consisted of carbon yarn sonication in a HNO_3_/H_2_SO_4_ oxidative solution for 15 min, followed by washing with distilled water until a pH of 6 is reached. 65% nitric acid (HNO_3_) and 95% sulfuric acid (H_2_SO_4_) were both purchased from Sigma-Aldrich. The oxidative solution was prepared according to [39], with a 1:3 HNO_3_/H_2_SO_4_ volume ratio. Yarns were dried at room temperature before testing.

### 2.3. Tensile and Pull-Out Tests

To evaluate the effectiveness of the different surface treatments on the mechanical properties of carbon yarns and on the bond with the inorganic matrix, a total of 60 tensile and 80 pull-out tests were carried out. Tensile tests on single carbon yarns were performed by using a tensile testing machine with a load bearing capacity of 50 kN, with a loading rate of 0.5 mm/min, according to ISO 10406-1 [40]. The FRP glass tabs were epoxy-bonded at the ends of the specimen to ensure a better grip during the test. A macro-extensometer with a gauge length of 50 mm was positioned at the center of each specimen to evaluate the elastic modulus and to measure the strain at failure (Figure 2a). Mechanical parameters have been reported as the average of five specimens for each type. Tensile strength has been calculated by dividing the tensile load by the cross-sectional area of the yarn (provided by the manufacturer). The elastic modulus has been calculated as the slope of the stress-strain curve in the elastic branch comprised of between 20% and 50% of the maximum tensile capacity [40].

Pull-out tests were carried out on carbon yarns embedded in a cubic specimen of cementitious mortar (40 × 40 × 40 mm^3^). The free length is kept constant and equal to 20 mm. Pull-out tests were performed using a tensile testing machine with a load bearing capacity of 5 kN. The specimen is fixed at the bottom by a metallic frame anchored to the testing machine, and the upper part of the yarn is gripped and pulled in displacement control at 0.5 mm/min (Figure 2b). 

### 2.4. Aging Conditioning Protocol

The same tests were carried out after subjecting the specimens to various artificial aging environments, as summarized in Table 3. In the case of pull-out specimens, the artificial conditioning started after curing the specimens for 90 days under laboratory conditions (20 ± 2 °C, RH = 70%).

The first environment (saline) comprises a 2.45% weight of sodium chloride (NaCl) and 0.41% weight of sodium sulphate (Na_2_SO_4_) aqueous solution. The concentration of NaCl and Na_2_SO_4_ was chosen according to ASTM D1141–98 [41]. In order to accelerate the aging process without promoting unrealistic chemical reactions, a temperature of 40 °C was chosen.

The second environment (alkaline) comprises a 4% weight sodium hydroxide (NaOH) aqueous solution with a pH of 13. The exposure to alkaline and saline environments was conducted by completely immersing the carbon yarns in the solution, while the pull-out specimens were immersed for 3 cm in order to keep the free length of the yarn out of the solution (Figure 3).

Freeze-thaw cycles consisted of freezing at −18 °C for more than 6 h and thawing at 40 °C for about 12 h. A total of 40 cycles were carried out both for carbon yarns and pull-out specimens. Finally, after artificial conditioning, all specimens were dried at 40 °C for 24 h before testing.

### 2.5. SEM and EDX Analysis

SEM and EDX analysis were carried out using a FESEM ZEISS SUPRA40 with an EDX-Detector Brucker Quantax 200-Z10, to investigate the atomic percentages of carbon, silicon, oxygen and on the surface morphology of carbon yarns after different surface treatments.

## 3. Experimental Results

### 3.1. Tensile Tests

The average tensile strength (σ_max_), ultimate strain (ε_u_) and elastic modulus (E) of carbon yarns with different surface treatments, subjected to different environmental exposures, are reported in Table 4, together with the corresponding coefficient of variation (CoV).

Carbon yarns impregnated with epoxy resin (C-E) showed the highest tensile strength, equal to 2327 MPa, which corresponds to an increase in the tensile strength of about 34%, if compared to untreated yarns. The impregnation with nano-silica dispersion also leads to a slight increase in the tensile strength, of about 10%, while apparently the oxidative treatment has barely influenced the mechanical properties of the yarn. The ability of organic and inorganic coatings to improve the mechanical properties of multi-filament yarns has been reported in different studies, and it is attributed to the stress transfer increase between single filaments [42,43]. The effectiveness of a coating to improve the yarn tensile strength depends on its ability to simultaneously engage the single filaments of the yarn during the test. This is in accordance with the results obtained by C-E_Ref and C-NS_Ref. Different failure modes can also be observed for treated carbon yarns. The C-Dry_Ref, C-NS_Ref and C-Ox_Ref yarns failed before all the carbon filaments had reached their maximum tensile strength (Figure 4a,c,d), suggesting that the nano-silica coating is not able to effectively activate all the yarn’s filaments during the tensile test. On the contrary, C-E_Ref yarns showed an abrupt and simultaneous breakage of all the yarns’ filaments (Figure 4b). These results confirm the superior ability of the epoxy resin to uniformly distribute the stress between the single filaments, thus increasing the yarn tensile strength. The elastic modulus seems not to be particularly affected by the surface treatment employed.

Looking at the results of durability tests (Figure 5), it can be observed that the tensile strength of carbon yarns is slightly affected by the exposure to saline and alkaline environments, regardless of the type of surface treatment applied. The most significant reduction in tensile strength was observed for the C-NS yarns. After immersion in the alkaline solution, the tensile strength of C-NS yarn showed a decrease in about 13%. Similar results were obtained after immersion in the saline solution, with a 12% decrease of the tensile strength. The nanosilica coating is therefore effective in increasing the yarn tensile performance in laboratory environmental conditions (C-NS), but it seems to suffer from exposure to saline and alkaline solutions. This effect was also found in other studies [44], where the ineffectiveness of nanosilica coatings was attributed not only to the reduced particle size (50 nm) but also to a partial wash out of the small particles when the reinforcement is immersed in the fresh cementitious paste washing. In this study, immersion of carbon yarns in saline and alkaline solutions at 40 °C could have caused this washout phenomenon. However, since SEM analysis of the yarn surface after exposure to various artificial environments has not yet been performed, these results will have to be confirmed by more detailed investigations.

### 3.2. Pull-Out Tests

The average maximum pull-out load and the total displacement corresponding to the maximum load (d_max_) are reported in Table 5, together with the corresponding coefficient of variation (CoV). Load-displacement curves are reported in Figure 6.

At first, it can be observed that all the surface treatments investigated in this study are able to increase the maximum pull-out load with respect to reference yarns.

Dry carbon yarns fail at low load values due to poor chemical-physical interaction between dry carbon filaments and the cementitious matrix. The failure is due to the breakage of some external filaments of the yarn and consequent slippage of the inner ones (telescopic effect). This failure mode can be observed in the broken specimen of Figure 7a, where only a few external filaments remained attached to the inorganic matrix after the pull-out test.

The use of epoxy-based coating proved to be the most effective treatment, by increasing the pull-out load of about 138% with respect to dry carbon yarns. This is due to the greater and more homogeneous stress distribution between all the filaments of the yarn (as also observed in tensile tests) and to the high friction which develops at the epoxy-to-inorganic matrix interface. These results are in agreement with some findings from the literature, which found an average 2–3 times increase in the mechanical performance of FRCM composites with polymer impregnated carbon yarns, compared to dry fiber yarns is observed [7,45]. Similar results have been found by Signorini et al. for epoxy coated glass fibers [13].

The maximum pull-out load of C-E samples occurred for very large displacements. In fact, once the yarn detaches from the inorganic matrix (first peak in Figure 6, C-E), it starts to slip within the matrix and the pull-out load increases until the maximum value is reached. This stress-hardening behavior can be explained by looking at the surface of the C-E yarns after pull-out (Figure 7b), which shows that the epoxy coating has been partially removed by friction with the inorganic matrix.

The maximum pull-out load of carbon yarns treated with oxidative solution (C-Ox) and nano-silica dispersion (C-NS) was respectively 28% and 40% higher than that of dry yarns (C-Dry). Looking at the load-displacement curves, C-Dry, C-NS and C-Ox specimens showed a similar pull-out behavior, characterized by a first linear increase in the load, followed by a quick load decrease after the peak. Since neither the nano-silica coating nor the oxidation treatment influenced the graph shape in the post-debonding region, it can be assumed that these treatments did not affect the frictional shear stress at the composite interface. Therefore, the higher peak loads obtained in both cases are the consequence of a higher chemical bond with the cementitious matrix. For C-NS yarns, this can be attributed to the chemical reaction of silica particles with the Ca(OH)_2_ of the cementitious mortar, forming a calcium silicate hydrate (C-SH) layer in the proximity of the fibers [26]. In regards to the C-Ox samples, the oxidation process is expected to modify the carbon fiber surface with the formation of oxygen-containing functional groups, which help the wettability of the fibers by the cementitious mortar [46]. Some studies in the literature show the effectiveness of different oxidative treatments in improving the bond strength between carbon fibers and cementitious mortars, and therefore also the mechanical properties of the composite. However, it is difficult to compare the results of this study to others from the literature due to many variables. Some studies only refer to short carbon fibers (instead of multifilament yarns) [46,47], while others use different oxidation processes or different setups for mechanical tests [48,49].

Regarding the oxidative treatment used in this study, some observations on its effectiveness are reported in Section 3.3, following SEM and EDX analysis on the yarn surface.

Regarding the results of pull-out tests after exposure to various artificial environments, a graphical representation of the outcomes is reported in Figure 8. It can be observed that, regardless of the environmental exposure, the unmodified yarns always show the lowest pull-out load. Moreover, the different environments do not seem to significantly affect the results of pull-out tests, regardless of the type of surface treatment employed.

It is interesting to note that C-E yarns subjected to alkaline and saline environments show a slight increase in the pull-out load, suggesting that the epoxy resin is able to further protect the carbon fibers from aggressive environments and that immersion in solution at 40 °C has even increased the bond at the epoxy-matrix interface. However, in some cases, a brittle failure at the yarn-matrix interface was observed, with the complete separation of the matrix into two parts (Figure 7c). This phenomenon can also be observed in some load-displacement curves, with an abrupt decay of the load corresponding to the matrix breakage (Figure 6, C-E_Sal, C-E_FT). This is confirmation that the superior properties of epoxy-coated carbon yarns are mainly due to the friction that develops during the slippage of the yarn within the matrix, rather than to the chemical adhesion between the cured epoxy resin and the matrix (very low). Therefore, the presence of an epoxy coating can act as a separating layer and promote delamination failures, as is also observed in other studies [13,50].

Surface treatments based on nano-silica (C-NS) and fiber oxidation (C-Ox) were shown to be adequately resistant to aggressive environments, confirming their superior properties compared to dry yarns. In this study, the most degrading environment was that of freeze-thaw cycles, which caused a decay of the pull-out load for all the specimens investigated (up to −27% for C-Ox specimens). This can be attributed to internal damage of the cementitious matrix as well as to an incomplete curing of the matrix due to the low temperature and humidity of the conditioning environment, which could have led to premature failure (although the matrix did not show any significant cracks due to freeze-thaw cycles).

### 3.3. EDX and SEM Analysis

The results of EDX analyses on the surfaces of C-Dry, C-NS and C-Ox yarns are reported in Table 6. The atomic percentage of carbon ranges from 83% up to 94%. The presence of oxygen atoms can be attributed to the organic sizing applied to carbon filaments during the manufacturing process of the fabric.

It is interesting to observe that EDX analyses conducted on C-Ox yarns excluded the formation of new oxygen-bearing groups on the carbon fiber surface (Table 6). A possible explanation of the improved interaction observed for C-Ox samples in pull-out tests is that the chemical treatment was not sufficient to promote the oxidation of the carbon backbone, but it was able to attack the fiber surface, causing an improvement of its roughness. An experimental study conducted on cement-based composites reinforced with carbon fibers (although short fibers) treated with concentrated acid, attributed the improved interaction between the matrix and the fibers to the formation of hydroxyl and carboxyl groups on the fiber surface [47]. However, in this study, the results of SEM analysis showed no significant difference in the surface morphology of the fibers before and after the treatment with the oxidative solution (Figure 9a,b). A further explanation is that no oxidation took place, but the acid was able to catalyze other chemical reactions which did not change the chemical composition of the fibers but could modify the oxygen bearing groups on the fiber surface. Acids are known to catalyze different chemical reactions, however, since the exact chemical composition of the sizing is unknown, it is not possible to state which phenomenon may have occurred. Moreover, because no clear FTIR spectrum of the fibers before and after chemical treatment could be acquired, this hypothesis cannot be confirmed.

On the other hand, the clear increase in the percentage of oxygen and silicon atoms observed for C-NS samples, compared to the untreated yarn, confirms the presence of nano-silica particles between the yarn filaments. However, SEM analysis (Figure 10) shows that the nano-silica coating is not uniformly distributed between the filaments, thus forming a discontinuous layer on the fiber surface. This is probably due to the manual impregnation process, which is not able to adequately control the uniformity of the application.

## 4. Conclusions

Based on the results of this experimental investigation, the following conclusions can be drawn:Pre-impregnation of multifilament carbon yarns with epoxy proved to be the most effective treatment, capable of increasing both the yarn tensile strength and the bond with the cement-based mortar. Epoxy was able to increase the carbon yarn tensile strength by about 34% and the pull-out load of about 138%. These effects, as well known from the literature, can be attributed to the ability of the low-viscosity epoxy to penetrate between single filaments of the yarn, thus guaranteeing a more homogeneous stress distribution through the yarn cross-section. However, it must be remembered that the use of organic polymers remains a weakness with regards to the mechanical behavior of the composite when exposed to high temperatures.The nano-silica coating was less effective than epoxy, but still able to increase the yarn tensile strength by 10% and the pull-out load by about 40%. However, the effectiveness of this treatment can be improved by optimizing the manufacturing process, to ensure a more homogeneous distribution of the particles on the yarn surface.The oxidation of carbon fibers with HNO_3_/H_2_SO_4_ solution seems not to substantially modify the mechanical properties of the carbon yarns. SEM analyses did not show significant changes in the surface of the carbon filaments after the oxidation process. However, this treatment was able to increase the pull-out load by about 28%. Further analyses are certainly needed to better investigate this aspect.Artificial aging in saline and alkaline environments caused only a slight reduction of the yarn tensile strength, which was always lower than 13%, regardless of the type of surface treatment applied.Pull-out tests carried out after exposure of the specimens in saline and alkaline environments showed no significant decrease in mechanical performance. Carbon yarns with epoxy impregnation showed the highest load values. Exposure to freeze-thaw cycles caused the greatest reduction in the pull-out load (between −10% and −27%), probably due to internal damage of the inorganic matrix (which in some cases broke in half during the test), rather than to deterioration of the carbon yarn.

## Figures and Tables

**Figure 1 materials-15-03927-f001:**
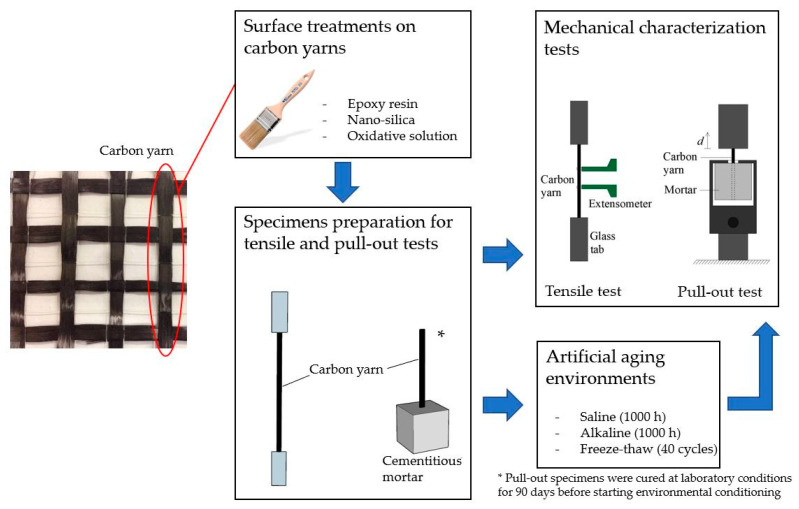
Schematic representation of the experimental campaign.

**Figure 2 materials-15-03927-f002:**
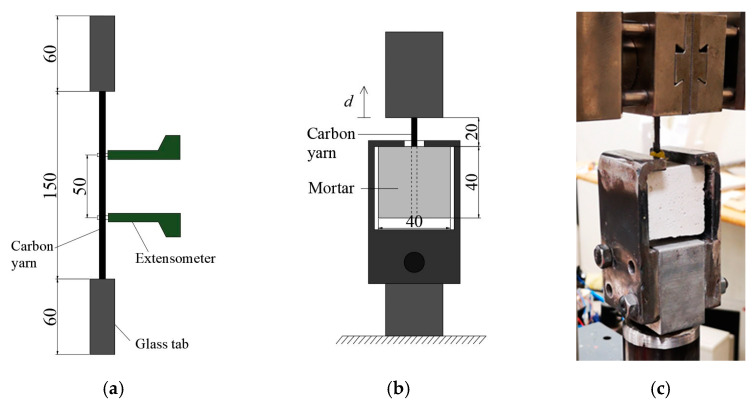
(**a**) Tensile and (**b**) pull-out test layout, (**c**) actual pull-out test setup.

**Figure 3 materials-15-03927-f003:**
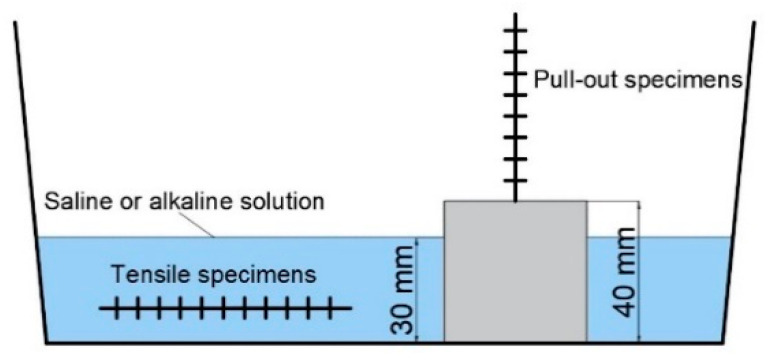
Conditioning exposure of tensile and pull-out specimens in saline and alkaline solutions.

**Figure 4 materials-15-03927-f004:**
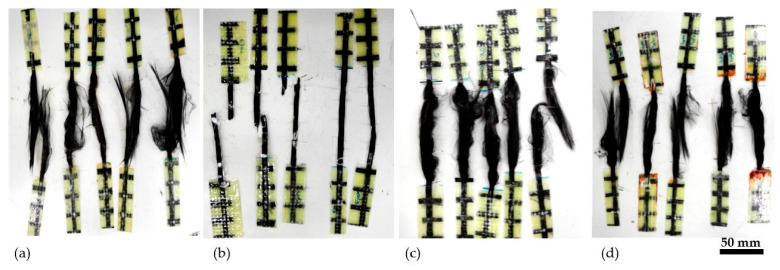
Failure modes: (**a**) C-Dry_Ref, (**b**) C-E-Ref, (**c**) C-NS_Ref, (**d**) C-Ox_Ref.

**Figure 5 materials-15-03927-f005:**
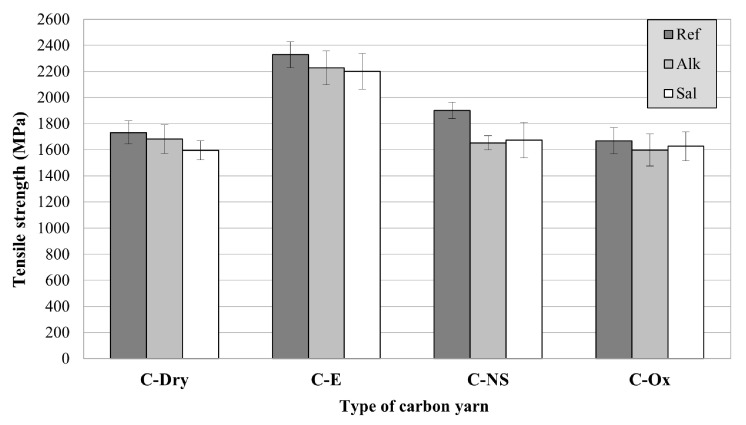
Tensile strength of carbon yarns with different surface treatments exposed to different environments.

**Figure 6 materials-15-03927-f006:**
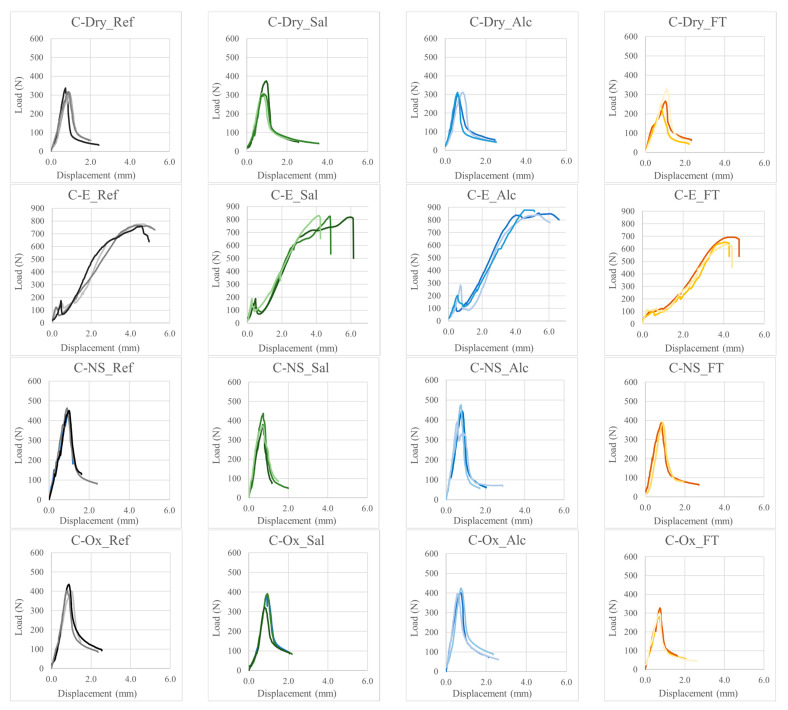
Load-displacement curves of pull-out tests.

**Figure 7 materials-15-03927-f007:**
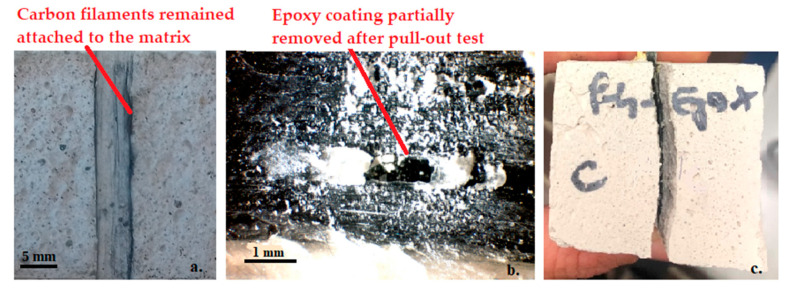
(**a**) Contact surface between dry carbon yarns (C-Dry) and inorganic matrix after pull-out tests, (**b**) external surface of epoxy coated carbon yarn (C-E) after pull-out, (**c**) matrix breakage.

**Figure 8 materials-15-03927-f008:**
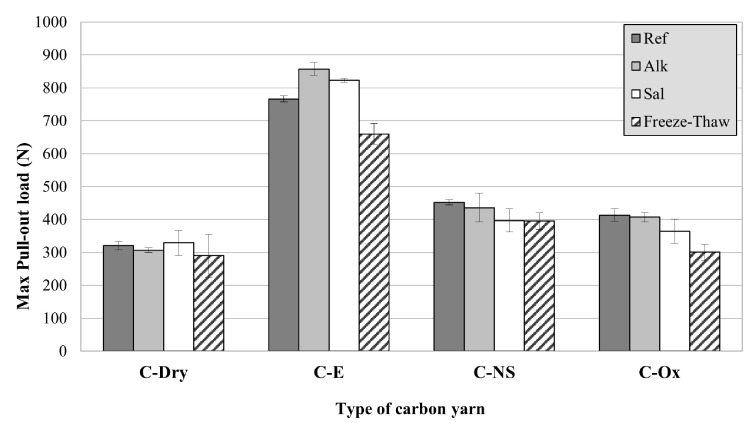
Pull-out load of carbon yarns with different surface treatments after artificial exposure.

**Figure 9 materials-15-03927-f009:**
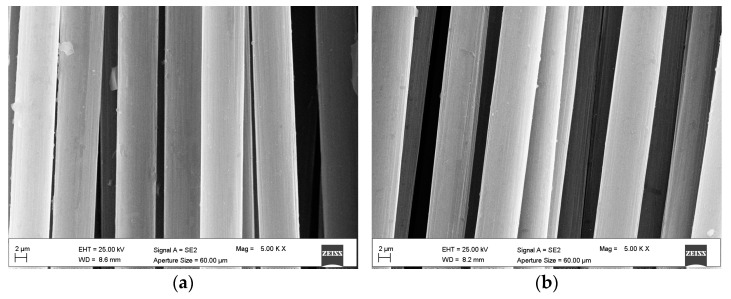
SEM images of (**a**) C-Dry, and (**b**) C-Ox multifilament yarns. Magnification 5.00 Kx.

**Figure 10 materials-15-03927-f010:**
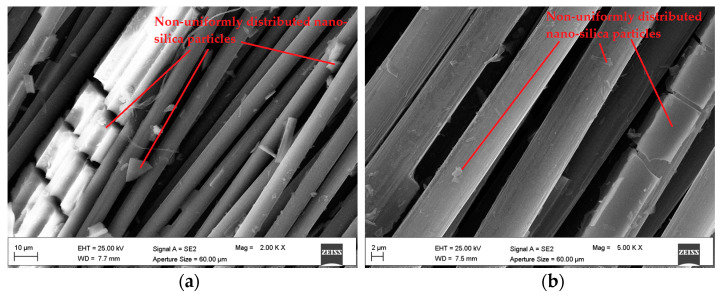
SEM images of C-NS yarns. Magnification (**a**) 2.00 Kx and (**b**) 5.00 Kx.

**Table 1 materials-15-03927-t001:** Mix proportions and mechanical properties of the inorganic matrix.

Material	CEM II/B-LL 32.5 R (kg/m^3^)	CEM II/B-LL 42.5 R (kg/m^3^)	CaCO_3_ 400 (kg/m^3^)	CaCO_3_ 600 (kg/m^3^)	Aerial Lime (kg/m^3^)	Water (kg/m^3^)	Compressive Strength (MPa)	Flexural Strength (MPa)
Cement-based mortar	82	165	715	205	110	260	17.95	5.66

**Table 2 materials-15-03927-t002:** Physical and mechanical properties of the epoxy resin (from manufacturer).

Material	Viscosity (mPa·s)	Tensile Strength (MPa)	Elongation at Break
Elan-tech EC 98N/W52	2000–3000	0.7–0.9	60–80%

**Table 3 materials-15-03927-t003:** Artificial aging test environments.

Environment	Temp	RH	Solution	Exposure Time	N° of Samples (5 for Each Surface Treatment)
None (Ref)	20 ± 2 °C	70%	-	-	20 tensile tests 20 pull-out tests
Saline	40 ± 2 °C	100%	2.45% NaCl + 0.41% Na_2_SO_4_	1000 h	20 tensile tests 20 pull-out tests
Alkaline	40 ± 2 °C	100%	4% NaOH	1000 h	20 tensile tests 20 pull-out tests
Freeze-Thaw	−18 ± 2 °C/+40 ± 2 °C	40%/100%	-	960 h (40 cycles)	20 pull-out tests

**Table 4 materials-15-03927-t004:** Results of tensile tests on carbon yarns.

Specimen	Environment		Tensile Strength σ_max_, (MPa)	Variation of Tensile Strength	Elastic Modulus E (GPa)	Ultimate Strain ε_u_ (%)
C-Dry	None (Ref)	Average	1732	-	145	1.39
		*CoV*	*6.9%*		*1.1%*	*1%*
	Saline	Average	1594	−8.0%	144	1.25
		*CoV*	*4.8%*		*4.7%*	*19.5%*
	Alkaline	Average	1681	−2.9%	152	1.07
		*CoV*	*10.1%*		*9.2%*	*5.8%*
C-E	None (Ref)	Average	2327	-	142	1.76%
		*CoV*	*7.6%*		*4.5%*	*5.1%*
	Saline	Average	2201	−5.4%	147	1.50
		*CoV*	*14.1%*		*5.2%*	*20.5%*
	Alkaline	Average	2226	−4.3%	151	1.47
		*CoV*	*11.4%*		*9.6%*	*17%*
C-NS	None (Ref)	Average	1900	-	143	1.6
		*CoV*	*3.3%*		*3.4%*	*6.1%*
	Saline	Average	1672	−12.0%	149	1.08
		*CoV*	*10.4%*		*3.4%*	*9.9%*
	Alkaline	Average	1651	−13.1%	150	1.24
		*CoV*	*3.3%*		*4.6%*	*12.3%*
C-Ox	None (Ref)	Average	1667	-	139	1.39
		*CoV*	*8.4%*		*4.3%*	*12%*
	Saline	Average	1626	−2.4%	155	1.10
		*CoV*	*9.7%*		*6.3%*	*15.1%*
	Alkaline	Average	1597	−4.2%	154	1.15
		*CoV*	*9.2%*		*12.6%*	*13.2%*

**Table 5 materials-15-03927-t005:** Results of pull-out tests of carbon yarns subjected to different aging protocols.

Specimen	Environment		Max Pull-Out Load (N)	Variation of Max Load (%)	Displacement at Max Load (mm)
C-Dry	None (Ref)	Average	321	-	0.86
		*CoV*	*4%*		*13%*
	Saline	Average	329	+2.5	0.87
		*CoV*	*11%*		*11%*
	Alkaline	Average	307	−4.4	0.69
		*CoV*	*2%*		*23%*
	Freeze-Thaw	Average	290	−9.7	0.96
		*CoV*	*23%*		*16%*
C-E	None (Ref)	Average	766	-	4.59
		*CoV*	*2%*		*5%*
	Saline	Average	823	+7.4	4.98
		*CoV*	*2%*		*17%*
	Alkaline	Average	857	+11.9	4.99
		*CoV*	*2%*		*12%*
	Freeze-Thaw	Average	660	−13.8	4.23
		*CoV*	*5%*		*6%*
C-NS	None (Ref)	Average	452	-	0.94
		*CoV*	*2%*		*7%*
	Saline	Average	397	−12.2	0.7
		*CoV*	*8%*		*3%*
	Alkaline	Average	436	−3.5	0.71
		*CoV*	*10%*		*19%*
	Freeze-Thaw	Average	395	−12.6	0.91
		*CoV*	*6%*		*11%*
C-Ox	None (Ref)	Average	413	-	0.94
		*CoV*	*5%*		*7%*
	Saline	Average	364	−11.9	0.94
		*CoV*	*10%*		*5%*
	Alkaline	Average	407	−1.5	0.78
		*CoV*	*5%*		*7%*
	Freeze-Thaw	Average	301	−27.1	0.74
		*CoV*	*8%*		*5%*

**Table 6 materials-15-03927-t006:** Atomic percentages of carbon, silicon and oxygen detected by EDX analyses on the surfaces of the yarns C-Dry, C-NS and C-Ox.

Sample	C (At %)	Si (At %)	O (At %)
C-Dry	93.47	0.23	6.31
C-NS	83.9	1.47	14.63
C-Ox	94.11	0.12	5.76

## Data Availability

Not applicable.

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
