# Peer review of "Fabric-Reinforced Cementitious Matrix (FRCM) Carbon Yarns with Different Surface Treatments Embedded in a Cementitious Mortar: Mechanical and Durability Studies"

_materials, 2022, doi:10.3390/ma15113927_

Round 1

Author Response

Dear reviewer,

thanks for your observations which made it possible to improve the quality of the manuscript.

Point-by-point replies are reported in the attached PDF file (Please see the attachment).

All changes made to the paper are highlighted in red in the revised document.

Reviewer 2 Report

This work by Bompadre et al. is nicely conducted and well organized. Thus, I fully support the publication of this work at Materials. Few minor comments may be considered by the authors: 1) In fig. 1, authors should remove the red line below the text. 2) Authors should re-phrase part of the conclusions section. Objective of this research is repeated in the conclusion part and in my view, this should be removed. The drawn conclusions should be re-phrased in a more concise way to deliver clear messages to readers. After authors make these edits, this paper could be accepted for publication. 

Author Response

This work by Bompadre et al. is nicely conducted and well organized. Thus, I fully support the publication of this work at Materials. Few minor comments may be considered by the authors:

1) In fig. 1, authors should remove the red line below the text.

Thank you for your observation. The red lines have been removed.

2) Authors should re-phrase part of the conclusions section. Objective of this research is repeated in the conclusion part and in my view, this should be removed. The drawn conclusions should be re-phrased in a more concise way to deliver clear messages to readers.

Thanks for this comment. Conclusions have been revised by removing the objective of the research and highlighting the experimental outcomes.

Reviewer 3 Report

1.     Brand and purities of the raw materials are missing.
2.     The value references must be cited for table 1.
3.      What is the function of Freeze-thaw cycles on the specimens?
4.       The Nano-silica was used to improve the mechanical priorities. What is the morphology of the nano silica as prepared by the Sol-Gel method?
5.       According to Fig.4, tensile strength of Ref samples is higher than other methods. That means that the surface treatment could not improve the T.S. In this case, how the authors can justify this matter.
6.      What is the magnification of Fig.5?

Author Response

1. Brand and purities of the raw materials are missing.

Brand and purities of the raw materials have been added.

2. The value references must be cited for table 1.

Values reference has been added in the description of Table 1 (Table 2 in the revised manuscript).

3. What is the function of Freeze-thaw cycles on the specimens?

Freeze-thaw cycles could affect the mechanical properties of the composite, in particular those of the inorganic matrix. Few studies have been performed regarding the effects of freeze-thaw cycles on the mechanical properties of FRCM systems. The results from the literature are often conflicting and difficult to compare, for this reason it is necessary to further investigate on this aspect.

4. The Nano-silica was used to improve the mechanical priorities. What is the morphology of the nano silica as prepared by the Sol-Gel method?

Unfortunately, we do not have detailed information on the morphology of the nano-silica. The only SEM images of the nano-silica particles are those shown in Figure 10.

5. According to Fig.4, tensile strength of Ref samples is higher than other methods. That means that the surface treatment could not improve the T.S. In this case, how the authors can justify this matter.

Tensile strength of Ref samples (C_Dry) is always lower than that of carbon yarns with surface treatments. The only method that does not particularly influence the tensile strength of the yarn is the oxidation process (C_Ox).

6. What is the magnification of Fig.5?

A scale has been added to Figure 5.

Reviewer 4 Report

The paper present a good topic related to FRCM carbon yarns with different surface treatments embedded in a cementitious mortar: mechanical and durability studies. The paper should be improved according to the comments in the attached file.

Author Response

(The authors gave the same response as above.)

Round 2

Reviewer 1 Report

The authors have improved the manuscripts as per the provided comments. Therefore, I have no additional comments.

Reviewer 4 Report

The authors responded to the comments. I recommend to accept the paper for publication